# Analysis of the Circulating miRNome Expression Profile in Saliva Samples After Neoadjuvant Chemoradiotherapy in a Rectal Cancer Study Population Using Next-Generation Sequencing

**DOI:** 10.3390/ijms262110506

**Published:** 2025-10-29

**Authors:** Kristóf Gál, Péter Dávid, Melinda Paholcsek, Márton Barabás, Endre Szilágyi, Krisztina Balogh, Dóra Solymosi, Szidónia Miklós, Johanna Mikáczó, Krisztina Trási, Emese Csiki, Mihály Simon, Péter Fauszt, Szilárd Póliska, Judit Remenyik, Árpád Kovács, Emese Szilágyi-Tolnai

**Affiliations:** 1Department of Oncoradiology, Faculty of Medicine, University of Debrecen, 4032 Debrecen, Hungary; gal.kristof@med.unideb.hu (K.G.); barabas.marton@med.unideb.hu (M.B.); balogh.krisztina02.24@gmail.com (K.B.); solymosi.dora@med.unideb.hu (D.S.); miklos.szidonia@med.unideb.hu (S.M.); mikaczo.johanna@med.unideb.hu (J.M.); trasi.krisztina@med.unideb.hu (K.T.); csiki.emese@med.unideb.hu (E.C.); simonm@med.unideb.hu (M.S.); 2Faculty of Agricultural and Food Sciences and Environmental Management, Complex Systems and Microbiome-Innovations Centre, University of Debrecen, 4032 Debrecen, Hungary; david.peter@agr.unideb.hu (P.D.); paholcsek.melinda@agr.unideb.hu (M.P.); szilagyi.endre@agr.unideb.hu (E.S.); fauszt.peter@agr.unideb.hu (P.F.); remenyik@agr.unideb.hu (J.R.); 3Department of Biochemistry and Molecular Biology, Faculty of Medicine, University of Debrecen, 4032 Debrecen, Hungary; poliska@med.unideb.hu

**Keywords:** miRNAs, rectal cancer, neoadjuvant concurrent chemoradiotherapy, tumor regression grade

## Abstract

Dysregulated microRNAs (miRNAs) have been implicated in the pathogenesis and progression of rectal adenocarcinoma. In this study, we aimed to identify miRNA alterations associated with the efficacy of neoadjuvant chemoradiotherapy in rectal cancer patients. High-throughput small RNA sequencing was performed to assess salivary miRNA expression profiles in 31 participants (11 rectal adenocarcinoma patients and 20 healthy volunteers). Paired saliva samples were collected from patients before and after chemoradiation. Tumor regression was classified according to the modified Ryan scheme into responders (tumor regression grade [TRG] 1–2, *n* = 10) and nonresponders (TRG3, *n* = 1). Bioinformatic integration of small non-coding RNA data revealed 37 miRNAs with distinct expression differences between patients and healthy controls. Furthermore, seven miRNAs showed significant alterations in response to radiotherapy. Among these, five candidates (hsa-miR-378a-3p, hsa-miR-203a-3p, hsa-miR-200a-5p, hsa-miR-361-5p, and hsa-miR-107) were successfully validated by RT-qPCR, displaying significantly increased salivary expression levels post-radiation compared with the pre-radiation samples (*p* < 0.05). Notably, hsa-miR-203a-3p, hsa-miR-200a-5p, and hsa-miR-361-5p demonstrated excellent discriminatory power for tumor regression grade (AUC > 0.7). Our findings support the involvement of specific salivary miRNAs in rectal adenocarcinoma tumor regression and highlight their potential as non-invasive biomarkers to evaluate treatment response following neoadjuvant chemoradiotherapy.

## 1. Introduction

Colorectal cancers (CRCs), due to their high incidence and the severity of the disease, represent a significant public health concern. Both in Europe and in Hungary, they are the second-most-common malignancy [1,2]. According to a 2020 survey, more than 10,000 new colorectal tumors are diagnosed annually in Hungary, of which more than one-third are rectal cancers [3]. Thanks to advances in therapeutic options, rectal cancer detected at an early stage is now highly treatable, with combined modalities—including systemic therapy, radiotherapy, and surgical interventions—offering substantial overall and disease-free survival benefits [4]. Based on the SEER database, the average five-year survival rate is 67%; for localized tumors, this figure reaches 90%, while even in cases of locoregional spread, it remains at 74% [5]. These values underscore the importance of early detection and screening programs.

One of the major therapeutic breakthroughs of the 2010s was the introduction of total neoadjuvant chemoradiotherapy. Several large international trials (RAPIDO, PRODIGE 23, and OPRA) have been conducted regarding TNT treatment protocols, demonstrating numerous advantages over conventional chemoradiotherapy regimens. Consequently, since 2021, TNT has been officially incorporated into international treatment guidelines [6,7]. Neoadjuvant radiotherapy constitutes an integral part of the TNT protocol [8]. Thanks to technological advances, pelvic radiotherapy is now delivered using intensity-modulated techniques with image guidance (typically cone-beam CT verification), enabling high precision and the administration of significant radiation doses while ensuring considerable protection of the surrounding healthy tissues [9,10].

The availability of different systemic protocol alternatives allows for individualized treatment tailored to tumor stage and patient condition. However, these approaches have inherent limitations, and thus, clinical outcomes remain highly heterogeneous [11,12]. This heterogeneity aligns with the molecular biological variability that has been increasingly explored and uncovered in CRC in recent years. Genetic mutations and pathological epigenetic alterations play crucial roles in tumor development and in the emergence of therapy resistance, making their comprehensive investigation and in-depth analysis indispensable [13,14,15].

The functional role of miRNAs has been gaining increasing prominence in scientific research, not only in tumor development and progression but also in therapeutic response and in the emergence of treatment-related toxicities. These small, non-coding, single-stranded RNAs play an essential role in post-transcriptional gene regulation, significantly influencing mRNA translation. Their role is undeniable in biological processes such as proliferation, cell differentiation, and apoptosis. The investigation of miRNAs offers opportunities for identifying biomarkers that may predict therapeutic efficacy or the occurrence of treatment-limiting toxicities, while also serving as potential candidates for targeted therapies [16].

In recent years, considerable attention has been directed toward alterations in microRNAs in response to cancer therapy. In breast cancer, Lindholm et al. identified miR-4465 as a potential biomarker during neoadjuvant bevacizumab and chemotherapy, showing that its downregulation effectively predicted reduced proliferation [17]. Based on the in vitro esophageal cancer study by Hummel et al., dysregulation of miR-27b-3p, miR-193b-3p, miR-192-5p, miR-378a-3p, miR-125a-5p, and miR-18a-3p was observed, consistent with their role in the negative post-transcriptional regulation of *KRAS*, *TYMS*, *ABCC3*, *CBL-B*, and *ERBB2* expression [18]. Yu et al. demonstrated that treatment of colorectal cancer cells with 5-FU led to the upregulation of 17 miRNAs (miR-19a, miR-20, miR-21, miR-23a, miR-25, miR-27a/b, miR-29a, miR-30e, miR-124b, miR-132, miR-133a, miR-141, miR-147, miR-151, miR-182, and miR-185) and the downregulation of three others (miR-200b, miR-210, and miR-224) [19]. Moreover, mir-21 was able to predict incomplete response in rectal adenocarcinoma after chemoradiotherapy [20]. According to Valenzuela et al., the miR-200 family is capable of reversing EMT phenotypes, thereby restoring chemosensitivity. Furthermore, miRNAs such as miR-19a and miR-625-3p have shown predictive value regarding chemotherapy response. Supporting this, Badr et al. reported that alterations in miR-19a-3p provide important insights into the long-term effects of 5-FU in colorectal cancer, particularly in relation to recurrence and disease progression [21]. Conversely, Chun-Ming Huan et al. found that miR-148a promotes apoptosis under chemoradiation while simultaneously suppressing proliferation in colorectal cancer cells. According to previous results, mir-345 was associated with unfavorable chemoradiotherapy response and poor locoregional control in rectal adenocarcinoma [22].

Numerous studies have investigated the relationship between radioresistance and alterations in miRNA expression. Research has suggested that miRNAs may modulate tumor radioresistance by influencing radiation-associated signaling pathways involved in DNA damage repair, cell cycle regulation, apoptosis, neovascularization, and inflammation. These pathways include EGFR, PI3K/AKT, NF-κB, RAS-MAPK, and TGF-β, as well as the JAK-STAT signaling cascade [23,24].

In the future, in addition to miRNAs, other non-coding RNAs, as well as markers of the tumor microenvironment and immune response, are expected to become key biomarker candidates, further refining personalized therapeutic approaches and enhancing the prediction of treatment outcomes. While many miRNAs have been proposed as potential biomarkers for forecasting responses to chemoradiotherapy (CRT) in rectal cancer patients, reliable and clinically validated markers for routine use are still lacking [25,26].

In this study saliva samples from patients with colorectal cancers were collected before the first and after the last radiotherapy treatment fractions and the miRNAs expression profiles were assessed using high-throughput small RNA sequencing. By integrating in silico bioinformatic analyses of small non-coding RNA data, it is possible to identify miRNAs exhibiting significant expression differences (*p* < 0.05) between the treated and control samples.

Based on these, our study pursued several objectives: (i) to examine distinctive miRNome signatures between healthy volunteers and patients with colorectal cancer, (ii) to determine whether there are any differentially expressed miRNAs in pre- and post-radiotherapy saliva samples, and (iii) to elucidate how miRNAs expression profiles could potentially provide objective information regarding the effects of radiotherapy based on tumor regression grade.

## 2. Results

### 2.1. Characteristics of the Study Cohort

In this prospective study, 31 participants (11 rectal carcinoma, 20 healthy controls) were recruited at the University of Debrecen, Faculty of Medicine, Department of Oncoradiology. The baseline demographics and clinical characteristics of the enrolled patients are shown in Table 1.

Participants in the cohort were balanced according to sex (six males and five females). The median age of the participants was 68. Patients were classified as responders and nonresponders according to the modified Ryan scheme classification: 90.91% were responders (TRG1-2) and 9.09% were nonresponders (TRG3).

### 2.2. General Description of Sequencing Results

The number of mapped cDNA reads was 4,164,594 ± 955,281 (75 bp each) per sample. The majority of the sequences were 21–23 nucleotides long. More than 90% of clean reads were retained after filtering out low-quality tags, removing adaptors, and cleaning up contaminants. Small RNA sequence types (represented by uniqueness) and length distribution were analyzed. Overall, more than 96% (±3%) of clean reads were assigned as miRNAs.

### 2.3. The Quantitative Evaluation of Small Non-Coding RNA Transcriptome Analyses Highlighted Both Overlapping and Cohort-Specific miRNAs

In-depth analysis of high-throughput small RNA transcriptome sequencing was performed to assess the miRNA expression profiles of our study population (13 participants), including 11 colorectal cancer patients and two sample pools from healthy volunteers. From the eleven patients with locally advanced rectal adenocarcinoma receiving concurrent neoadjuvant chemoradiotherapy, we collected saliva samples before the first (initial phase) and immediately after the last (finishing phase) radiotherapy treatment fractions. Overall, 2882 miRNAs were identified, and 2728 miRNAs were omitted from the analysis due to a very low read number (read per million [RPM < 10]) in all the samples. With this, we found 154 miRNAs having a read number above or equal to 10 (RPM ≥ 10) in the examined samples. For further analysis we assessed this set of miRNAs.

The number of uniquely and commonly expressed miRNAs (i.e., intersections) is illustrated in the Venn diagram (Figure 1A). The miRNA transcriptome profiles showed notable differences between the experimental groups. Overall, 147 miRNAs were consistently detected across all groups. We detected that seven miRNAs (hsa-miR-576-5p, hsa-miR-106b-5p, hsa-miR-29b-3p, hsa-miR-548e-3p, hsa-miR-3614-5p, hsa-miR-221-5p, and hsa-miR-505-3p) were uniquely expressed in our patients population (initial and finishing phase). Interestingly, from these seven miRNAs, only the hsa-miR-221-5 was significantly upregulated in the finishing phase compared to the initial phase.

Circo plots representing distribution profiles of the core 154 miRNAs (Figure 1B) exhibit remarkable expression changes when comparing healthy controls to colorectal patients either before the first (initial phase) and last (finishing phase) treatment fractions.

### 2.4. miRNA Expression Profiling in Rectal Cancer: Differential Signatures Associated with Disease Status and Radiotherapy Response

Differential expression analysis was performed to identify the distinctly expressed miRNAs between the rectal cancer study population and healthy control groups (Figure 2A). For these analyses we used the 154 miRNAs with RPM levels above 10 (RPM ≥ 10) from any of the samples. As the volcano plot indicates, 37 miRNAs showed statistically significant expression differences between the healthy controls and the rectal cancer study groups. From these, 16 miRNAs exhibited reduced expression levels while 21 miRNAs showed increased expression levels relative to the control groups. We investigated the effects of radiotherapy on the miRNAs expression profiles of rectal cancer patients (Figure 2B). The expression differences in miRNAs with RPM levels above or equal to 10 (RPM ≥ 10) were assessed for these analyses between the initial and the finishing phase. According to our results, seven miRNAS (hsa-miR-378a-3p, hsa-miR-203a-3p, hsa-miR-200a-5p, hsa-miR-152-3p, hsa-miR-361-5p, hsa-miR-107p, and hsa-miR-221-5p) exhibited remarkable expression differences between the initial and the finishing phase, all of which were characterized by overexpression. From these miRNAS, the hsa-miR-378a-3p, hsa-miR-203a-3p, hsa-miR-200a-5p, and hsa-miR-152-3p were also found as a differentially expressed miRNAs when comparing the control and the patient groups. Notably, only one miRNA (hsa-miR-221-5p) reflected the Venn diagram results and showed significant overexpression differences caused by the radiotherapy.

Ordination pattern analysis of the differentially expressed miRNAs was performed to identify distinct clusters among the experimental groups (Figure 2C,D). Beta diversity relationships are visualized in two-dimensional scatterplots, where each point represents an individual patient sample. The distances between points reflect differences in miRNA expression profiles.

Notably, distance-based dissimilarity matrices indicated that neither the comparison between healthy individuals and rectal cancer patients (Figure 2C), nor the comparison between the initial and final phases, revealed substantial effects on overall community variation (Figure 2D). As a result, no distinct or non-overlapping clusters were observed.

A cluster heatmap was created to represent the expression levels of all miRNAs with RPM ≥ 10 values (Figure 2E).

### 2.5. Validation of Differentially Expressed miRNAs by RT-qPCR (Reverse Transcription Quantitative Polymerase Chain Reaction)

To confirm the consistent gene expression tendencies seen in our miRNA-seq data, seven significantly altered miRNAs caused by the radiotherapy (Figure 3A) were tested via RT-qPCR as well. The relative expression changes were calculated according to Livac’s 2^−ΔΔCT^ method (Figure 3B).

According to our results, five miRNAs (hsa-miR-378a-3p, hsa-miR-203a-3p, hsa-miR-200a-5p, hsa-miR-361-5p, and hsa-miR-107p) were validated successfully by RT-qPCR across our sample groups (Figure 3B). Of note is that these expression patterns were consistent with the RNA sequencing results (Figure 3A).

Among these five miRNAs, all of them were upregulated compared to the initial phase. As a result, five significantly altered miRNAs (hsa-miR-378a-3p, hsa-miR-203a-3p, hsa-miR-200a-5p, hsa-miR-361-5p, and hsa-miR-107p) were identified as candidates for further testing. In the cases of hsa-miR-152-3p and hsa-miR-221-5p, we observed similar expression patterns with the RNA sequencing results; nevertheless, the expression differences were not significant between the initial and the finishing phases.

According to these results, we wondered whether these five miRNAs could predict the success of therapy based on the fact that radiotherapy significantly increased their levels. To answer this question, ROC analyses were performed to calculate the predictive potential of the miRNAs.

### 2.6. Prognostic Performance of miRNA Biomarkers from Saliva Samples Derived from Colon Cancer Patients

The first approach to identifying molecular markers capable of predicting tumor response to neoadjuvant radiotherapy was to estimate the capabilities of miRNAs to discriminate IP and FP stages from the saliva samples. For this purpose, receiver operating characteristic (ROC) curve analysis was applied to discriminate between the miRNA expression patterns in the initial (IP) and finishing phases (FP), using relative gene expression data obtained from RT-qPCR (Figure 4A). We calculated the area under the curve (AUC) for each miRNA.

On the basis of qRT-PCR-validated gene expression analyses, two miRNAs (hsa-miR-378a-3p (AUC = 0.87) and hsa-miR-203a-3p (AUC = 0.84)) demonstrated excellent discriminatory power. Moreover, hsa-miR-200a-5p and hsa-miR-361 were found to display high diagnostic potential with AUC > 0.7. Based on our preliminary results, hsa-miR-378a-3p, hsa-miR-203a-3p, hsa-miR-200a-5p, and hsa-miR-361 have the promising potential to discriminate the initial phase from the finishing phase, but further validations are needed. In the case of hsa-miR-107-5p, we observed lower diagnostic potential compared to the other miRNAs—the AUC value was 0.55.

In addition, the relative gene expression values of patients (IP and FP) were also dichotomized by plotting the sensitivity values in relation to 1-specificity to estimate the optimal cutoff values for these biomarkers (Figure 4B). This approach was used to determine the optimal cutoff values for these biomarkers. We observed that the hsa-miR-378a-3p displayed 1.03, hsa-miR-203a-3p showed 0.85, hsa-miR-200a-5p had 1.56, hsa-miR-361-5p displayed 0.45, and hsa-miR-107 showed 2.98 cutoff values.

After determining the significantly altered miRNAs’ differential potentials between the IP and FP phases, we evaluated their efficiency in discriminating treatment responses based on tumor regression grade (TRG1, TRG2, and TRG3) (Figure 5). In order to estimate their efficiency in correctly classifying patients according to tumor regression grade, we calculated AUC values for significantly altered miRNAs using ROC analyses.

In this analysis, the relative miRNA expression levels of patients in the TRG1 and TRG2 groups were compared with the TRG3 tumor regression group. According to our results, hsa-miR-378a-3p exhibited poor discriminatory power with a 0.6 AUC value and 1.2 cutoff value (Figure 5A). In the case of hsa-miR-203a-3p, high diagnostic performance was observed with a 0.98 AUC and 0.74 cutoff point (Figure 5B). hsa-miR-200a-3p showed an acceptable discriminating power, with an AUC of 0.78 and a 1.61 cutoff value (Figure 5C). Additionally, hsa-miR-361 was also found to display an acceptable diagnostic potential with a 0.78 AUC value and a 1.61 cutoff value (Figure 5D).

Moreover, hsa-miR-107 exerted only a weak differential potential with a 0.61 AUC value regarding tumor regression (Figure 5E).

These results suggest that hsa-miR-203a-3p, in combination with hsa-miR-200a-3p and hsa-miR-361, could be a putative predictive marker for tumor regression, but further validations are needed. It is important to note that our findings are preliminary observations and because of the small sample and unbalanced sample size, external validations in larger, independent cohorts are needed to ensure the reproducibility and generalizability of the findings.

### 2.7. Biological Functions Affected by Dysregulated miRNAs in CR Patients

miRNAs influence several biological functions. In order to assess the relevant biological pathways that are affected by dysregulated miRNAs in CR pathogenesis, the in silico data prediction tool miRabel and KEGG pathway databases were used together (Figure 6). This algorithm considers evolutionary conservation, Watson–Crick base pairing, and the thermodynamic properties between the miRNA seed region and its target mRNA [27]. miRabel scores were estimated to grade miRNA/mRNA interactions, with a 0.05 threshold suggested by the developers. KEEG database prediction pathway analyses revealed that the miRNA target genes were enriched in several important biological functions, including cell ECM adhesion, immune response, pathways in cancer, HIF-1 α signaling pathway, cell adhesion, autophagy, apoptosis, cell cycle, tight junction, metabolic pathways, and IBD pathogenesis.

Based on miRabel, together with the KEGG pathway, we found that hsa-miR-378-3p, hsa-miR-203a-3p, and has-miR-361-5p regulate genes that are highly involved in cell ECM adhesion (fibronectin, *MMPs*, and *TIMP*), immune response (Caspase-9, *MALAT1,* and *NEAT1*), and apoptosis (*KLK4*, *FOXD1-MYC-ODC1*, and *STAT3*). hsa-miR-200a-5p mostly influenced the genes that are involved in autophagy (*PDK1*, *Akt/mTOR*, and *ULK1*) and cell adhesion (*MMP3*, *MINK1*, and *LAMA4*) pathways. Furthermore, hsa-miR-107 (*ATG12*, *LC3-II*, and *HMGB1*) mainly exacerbated autophagy mechanisms, apoptosis (*BCL2* and *PTEN/AKT*), and immune response (*SOCS3*/*STAT3*) pathways.

## 3. Discussion

Colorectal cancer (CR) is a disorder that involves the unregulated proliferation of glandular epithelial cells in the colon or rectum. The high mortality rate associated with CR has driven researchers to find ways for new innovative diagnostic and treatment strategies [28]. Among these, miRNAs have gained attention in relation to cancer diseases based on their crucial role in regulating several biological processes that implement the initiation, progression, and metastasis of CR. Accordingly, altered expressions of miRNAs have been investigated and have confirmed their regulatory role in a wide variety of cancer types. Various non-coding, small RNA segments can act as oncogenes or tumor suppressors, thus influencing the progression of tumor processes. Although several miRNAs have been described as CRC-specific miRNA signatures, their biomarker effects in relation to therapy efficiency have only been partially confirmed [25].

In the present study, we evaluated the distinctive miRNA signatures between healthy volunteers and patients with colorectal cancer. We wanted to assess those miRNAs whose expression levels were significantly altered by radiotherapy. Moreover, our aim was to investigate how miRNAs profiles are associated with treatment response, motivating the search for their role as a prognostic molecular marker in CR.

Saliva-based liquid biopsy and the analysis of miRNA expressions in saliva samples are not new procedures. Their significant advantage is that, compared to venous blood sampling, they are a non-invasive procedure that can be repeated regularly and painlessly. Previous studies have shown that 20–30% of the proteins in saliva and blood are identical, indicating a biological connection between blood and saliva. This suggests that extracellular vesicles from the tumor may reach the salivary glands through circulation, where they can be taken up by the acini through endocytosis [29,30,31]. In addition to active secretion, passive leakage, either resulting from tissue necrosis or apoptosis, plays a significant role [32]. The miRNAs that enter the circulation in this way have different forms and different stabilities. While naked miRNAs are extremely unstable, and therefore sensitive to degradation by nucleases, proteins, especially exosomal miRNAs, show greater stability [33]. Exosomal miRNAs packaged in a lipid bilayer are excellent biomarker candidates due to their stability [34]. It should be emphasized that these formulas are typically formed by active secretion, thus an objective picture of the physiology of tumor processes can be obtained [35].

We performed small transcriptome sequencing on the saliva samples of CR patients and healthy volunteers. Based on our results, distinctive differences in miRNAs expression profiles among study groups were found. Notably, 37 miRNAs were significantly altered between the CR and healthy control groups. Furthermore, seven miRNAs (hsa-miR-378a-3p, hsa-miR-203a-3p, hsa-miR-200a-5p, hsa-miR-152-3p, hsa-miR-361-5p, hsa-miR-107p, and hsa-miR-221-5p) exhibited significant expression differences caused by radiotherapy. Interestingly, all of the seven miRNAs expression levels increased after the last treatment fraction. This may suggest that in CR pathogenesis, important regulatory mechanisms are malfunctioning due to the decreased level of these important miRNAs. Taken together, these considerations lead us to hypothesize that the expression levels of the aforementioned miRNAs may notably be associated with the success of radiotherapy treatment.

After PCR validation, we were able to identify five miRNAs (hsa-miR-378a-3p [*p* < 0.005], hsa-miR-203a-3p [*p* < 0.005], hsa-miR-200a-5p [*p* < 0.001], hsa-miR-361-5p [*p* < 0.05], and hsa-miR-107-3p [*p* < 0.05]) that had statistically increased expression between the initial and the finishing treatment phases. All of these miRNAs have been described in previous studies examining the expression of miRNAs in CR blood and colon biopsy samples.

A previous report shows that hsa-miR-378a-3p was downregulated in CR patients [36]. Furthermore, it plays a notable role in carcinogenesis, exhibiting tumor-suppressive activity. hsa-miR-378a-3p also regulates those genes that are highly involved in apoptosis (*KLK4*, *FOXD1-MYC-ODC1*, and *STAT3*) and cell cycle arrest (*CCND1*) and reduce the proliferation (*CDK1*) and migration of CR cells. CR patients with lower hsa-miR-378a wereassociated with a significantly shorter survival time than patients with higher hsa-miR-378a [37]. Restoration of hsa-miR-378a-3p limits the development of tumor cell lines and reduces tumor growth, according to mouse xenograft studies. hsa-miR-378a-3p also exacerbated significant modulatory and silencing effects on the expression of *MALAT1* and *NEAT1*, which are well known to influence tumor progression and inflammatory processes through the propagation of the NF-κB pathway [38,39]. Based on these findings, the overexpression of miR-378a (or restoring its expression) could suppress proliferation and may help slow tumor growth. Consistent with previous studies, we found significantly higher expression (*p* < 0.005) of hsa-miR-378a-3p in saliva samples after the last treatment fraction of neoadjuvant chemoradiotherapy compared with the initial phase. However, only a poor discriminatory potential (AUC = 0.6) was observed with this miRNA regarding tumor regression.

According to the literature, hsa-miR-203a-3p is often downregulated in CR tissues compared to adjacent normal tissues, suggesting that hsa-miR-203a-3p might be a tumor suppressor gene in CR. Its overexpression hindered tumor invasion by inhibiting epithelial–mesenchymal transition and increased the apoptotic rate of tumor cells. Furthermore, a negative correlation was found between the expression of *THBS2* and hsa-miR-203a-3p; thus, *THBS2* was considered as a target gene responsible for tumor progression and angiogenesis [40]. Moreover, hsa-miR-203a-3p downregulates IGF-1R receptors involved in cell survival and proliferation [41]. In contrast to these findings, Chen et al. demonstrated the overexpression of hsa-miR-203a-3p in colorectal tumor patients and described its tumor-promoting effect through the downregulation of *PDE4D* [42]. Although previous results are contradictory, our study shows that in the saliva samples of patients who respond well to therapy and show regression, the expression of hsa-miR-203a-3p increases significantly (*p* < 0.005) after neoadjuvant radiochemotherapy and has actable prognostic performance. It has diagnostic potential based on the difference in expression between the healthy control groups and the patient population.

Previous studies have shown that low hsa-miR-200a expression significantly worsens the prognosis of patients, and it has been associated with radioresistance, tumor aggressiveness, and poor regression grades in patients with CR. It plays an important role in the pathogenesis of several tumor types by targeting *ZEB1* and *ZEB2*, which are transcriptional repressors of E-cadherin, thereby maintaining epithelial integrity [43]. Decreased level of miR-200a facilitates epithelial–mesenchymal transition (EMT), which causes a loss of cell adhesion and increased invasion potential. According to previous studies, *SIX1* gene expression is remarkably associated with EMT. Overexpression of the *SIX1* gene inhibits CDH-1 expression, promoting EMT in CRC. *SIX1*-induced *CDH1* repression and *EMT* were partly related to the post-transcriptional activation of *ZEB1* and the transcriptional inhibition of the miR-200 family [44]. In the present study, neoadjuvant therapy increases hsa-miR-200a (*p* < 0.001) expression in the saliva of tumor patients, and it has high prognostic and diagnostic potential. Based on clinical data and prognostic performance studies, hsa-miR-378a-3p has excellent biomarker potential for monitoring the efficacy of treatment. Its expression also differed significantly between the patient and control samples, so it may be suitable for diagnostic purposes.

*SND1* is a multifunctional protein that plays an important role in carcinogenesis. Its overexpression has been shown in prostate, breast, and colon tumors, especially in tumors with a high metastatic rate. It can regulate intracellular processes such as RNA interference as a nuclease, mRNA splicing, and transcription as a coactivator. In mouse studies, SND1 has been shown to promote tumor angiogenesis by activating NF-κB and inducing anigogenin. Based on the results of Ma F et al., hsa-miR-361-5p can directly bind to the 3′-UTR regions of *SND1*, thereby significantly reducing gene expression [45]. hsa-miR-361-5p has also been identified as a tumor suppressor in other tumor types. Its increased expression improves progression-free survival in breast cancer and can inhibit carcinogenesis in HCC by targeting Twist1 [45]. In contrast, other studies have described the overexpression of hsa-miR-361-5p in ovarian tumors and emphasized its potential oncogenic role in lung adenocarcinoma [46,47]. Based on our results, its expression in saliva is significantly (*p *< 0.05) increased after radiochemotherapy of rectal adenocarcinoma, and it has high prognostic potential.

Emerging evidence suggests that low miR-361-5p expression may contribute to reduced radiosensitivity in rectal adenocarcinoma, making it a candidate biomarker for predicting TRG and guiding personalized therapy. miR-361-5p regulates a wide spectrum of oncogenic pathways. It targets *VEGF*, *TWIST1*, and *STAT6*, thereby influencing angiogenesis, EMT, and immune modulation [48,49,50]. Therefore, loss of miR-361-5p expression promotes pro-survival signaling and invasive phenotypes, features often associated with resistance to radiotherapy. Experimental data show that miR-361-5p negatively regulates anti-apoptotic genes such as *BCL2* [48]. Downregulation of miR-361-5p may thus enhance cellular survival after radiation-induced DNA double-strand breaks, allowing malignant clones to persist.

hsa-miR-107 has previously been identified as a tumor suppressor in hepatocellular carcinoma, lung carcinoma, melanoma, and squamous cell carcinoma [51]. In hepatocellular carcinoma, it can increase tumor cell apoptosis by regulating chemosensitivity. By targeting cofilin-1, it promotes the accumulation of ROS in cells. However, according to the study of Liang Y et al., it can induce chemoresistance by targeting calcium-binding proteins in colon tumors, so it may be a potential target molecule during systemic treatments [52]. Based on our results, its expression in saliva increases significantly (*p* < 0.05) after neoadjuvant treatment, but its prognostic significance is low based on performance studies.

Among them, hsa-miR-107 has emerged as a multifunctional miRNA with context-dependent tumor-suppressive or oncogenic roles. In gastrointestinal cancers, including rectal adenocarcinoma, low miR-107 expression has been linked to tumor progression, therapy resistance, and poor prognosis. Recent studies suggest that reduced miR-107 may impair radiosensitivity, making it a candidate predictor of nCRT efficacy [53]. miR-107 targets several cell cycle regulators, including *CDK6* and *Notch2*, thereby influencing G1/S transition [54]. Loss of miR-107 results in unchecked proliferation, which may enable tumor cells to rapidly repopulate after radiotherapy-induced injury. Moreover, in colorectal adenocarcinoma, miR-107 overexpression has been strongly associated with the regulation of specific target genes that influence tumor progression, such as transferrin receptor 1 (*TFR1*). Furthermore, miR-107 was found to modulate tumor angiogenesis through the regulation of hypoxia-inducible factor 1β (*HIF-1β*), leading to decreased vascular endothelial growth factor (*VEGF*) expression and suppressed tumor growth [54].

During our research, the expressions of two miRNAs were increased in the saliva of CRC patients (hsa-miR-152-3p and hsa-miR-221-5p), which we could not validate by PCR analysis. hsa-miR-152-3p has been identified by several publications as a tumor suppressor in ovarian, endometrial, and breast tumors, as well as in prostate tumors [55,56]. However, in CRC patients, its high expression negatively regulates the gene expression of *AQP11*, which promotes cell adhesion and hepatic metastases [26]. The expression of hsa-miR-221-5p was clearly increased in CRC patients as compared to the healthy population, and it was identified as a potential oncogene. Interestingly, its expression in saliva increased after treatment, whereas previous studies have shown that it clearly decreased in plasma after radiation therapy [57]. Conversely, it functions as a tumor suppressor in prostate cancer [58]. Although both miRNAs appeared to be differentially expressed in sequencing analysis, these results could not be consistently validated by qRT-PCR. This discrepancy highlights the methodological differences and potential technical limitations of both approaches. Discrepancies between NGS and qRT-PCR results may arise from several factors. From a technological perspective, NGS is highly sensitive and covers a broad dynamic range, but it may generate false positives for rare or low-abundance miRNAs. In contrast, qRT-PCR is targeted and accurate, yet often less sensitive to certain sequences, such as GC-rich regions or those forming stable secondary structures. NGS outcomes are also influenced by the choice of mapping algorithms, normalization methods, and cutoff thresholds. Furthermore, sequencing and PCR are often performed on non-identical sample subsets, where even minor biological variations can lead to inconsistent findings. Therefore, results from candidates should be interpreted with caution and treated as preliminary observations that require further confirmation.

Based on our findings, we suggest that hsa-miR-203a-3p in combination with hsa-miR-200a-3p and hsa-miR-361 could be a putative predictive marker for tumor regression, but further validations in larger and independent cohorts are needed to confirm the robustness and generalizability of the proposed biomarkers.

We also performed pathway analysis of the KEGG database. According to the in silico pathway analyses, eleven relevant biological pathways, including cell ECM adhesion, immune response, pathways in cancer, HIF-1 α signaling pathway, cell adhesion, autophagy, apoptosis, cell cycle, tight junction, and metabolic pathways, were identified to be influenced by dysregulated miRNAs in CR patients.

Our study has several limitations. First, the aforementioned low and unbalanced sample size might have contributed to the resulting excellent diagnostic potential in the case of hsa-miR-203a-3p, hsa-miR-200a-3p, and hsa-miR-361. We acknowledge that the relatively small sample size may have influenced the ROC values; therefore, with a larger cohort, these values might be lower than those observed in the present study. Second, miRNAs have a pleiotropic effect; they can impact their gen regulatory impacts on several biological pathways. Therefore, it is difficult to determine which biological pathways were influenced by the altered expression of the proposed miRNAs and how these changes relate to treatment response following neoadjuvant chemoradiotherapy. Continued research and validation are essential to draw far-reaching conclusions on whether and how miRNAs contribute to CR pathology and to support their differential diagnostic potential.

In summary, our study provides novel insights into the potential applicability of miRNAs as differential prognostic markers of CR in terms of radiotherapy success that may pave the way for future development of non-invasive and cost-effective diagnostic and therapy monitoring tools for CR.

## 4. Materials and Methods

### 4.1. Clinical Methods and Study Cohort

Patient recruitment for the study was conducted at the Department of Oncoradiology of the Clinical Center of the University of Debrecen, under institutional ethics committee approval. All enrolled patients had histopathologically confirmed rectal adenocarcinoma and were selected for consolidation total neoadjuvant therapy (TNT) based on multidisciplinary tumor board recommendations. In addition to a diagnosis of locally advanced rectal adenocarcinoma, adequate patient compliance, and the provision of voluntary, informed consent were mandatory inclusion criteria. Age and sex were not considered determining factors for patient inclusion.

Exclusion criteria encompassed all physiological conditions that contraindicated either chemotherapy or radiotherapy, as well as comorbidities with the potential to significantly influence miRNA expression. Consequently, only patients with an overall condition suitable for neoadjuvant chemoradiotherapy (ECOG performance status 0–2) were considered eligible. Further exclusion criteria included the presence of any additional malignant disease, a history of radiotherapy to any anatomical region, or prior chemotherapy or the presence of distant metastases. None of the patients presented with active oral or systemic infections, immunological disorders, or contraindications such as bone marrow failure, anemia, leukopenia, thrombocytopenia, untreated hepatic or renal impairment, or cardiac insufficiency. Patients with regular alcohol or drug consumption or active smoking habits were also excluded.

With respect to comorbidities, essential hypertension was the most common, observed in 8 patients (72.7%). In addition, one patient had non-insulin-dependent diabetes mellitus, one suffered from tachycardia, and another had a history of nephrolithiasis.

Radiotherapy was delivered to a total dose of 50.4 Gy, administered in daily fractions of 1.8 Gy, five days per week (Monday through Friday). Concurrently, patients received weekly bolus infusions of 5-fluorouracil (5-FU) at a dose of 500 mg/m^2^ as radiosensitization during the first and fifth weeks of radiotherapy (Table 2).

Localization CT scans of the pelvis were acquired prior to treatment using non-contrast-enhanced imaging with a slice thickness of 3 mm on a SOMATOM go.Sim (Siemens Healthineers AG, Forchheim, Germany). Target volume delineation, including gross tumor volume (GTV), clinical target volume (CTV), and planning target volume (PTV), was performed in accordance with the RTOG Contouring Atlas guidelines. Radiotherapy planning and contouring were executed using the Raystation treatment planning system (RaySearch Laboratories AB, Stockholm, Sweden). Target volume definition was refined through multimodal image fusion incorporating MRI and PET-CT datasets for optimal anatomical and metabolic accuracy.

Treatment plan quality assurance was conducted via dose–volume histogram (DVH) analysis, with dosimetric constraints for organs-at-risk (OARs) evaluated in compliance with QUANTEC (Quantitative Analyses of Normal Tissue Effects in the Clinic) criteria [59]. Radiotherapy was administered using Elekta Versa HD linear accelerators with 6 MV photon beams, employing volumetric modulated arc therapy (VMAT) delivery techniques.

During the initial treatment week, daily cone-beam computed tomography (CBCT) image guidance was performed for setup verification, followed by adaptive correction averaging on treatment day six. Subsequent fractions were verified weekly utilizing CBCT imaging for ongoing positional accuracy.

Following chemoradiotherapy, patients received additional chemotherapy according to a three-month FOLFOX or XELOX protocol, which was subsequently followed by restaging imaging. Restaging evaluations included high-resolution pelvic MRI and/or PET-CT imaging, along with proctological assessment and endoscopic evaluation, to determine radiologic and clinical response and to guide subsequent management decisions. Imaging re-evaluation was performed within one week after the last cycle of consolidation chemotherapy.

Tumor regression grade was assessed using the modified Ryan scheme and categorized as follows: TRG1 indicates regression with single cells or little groups of cancer cells (near complete response), TRG2 reflects residual cancer with evident tumor regression but more than single cells or rare small groups of cancer cells (partial response), and TRG3 corresponds to extensive residual cancer with no evident tumor regression (poor or no response).

### 4.2. Sampling and RNA Extraction

Saliva samples from colorectal cancer patients and healthy volunteers were collected in 1 mL nuclease-free collection tubes (Screw cap microtube, Sarstedt, Lower Saxony, Germany, Cat. 72.694.700) containing DNA/RNA transport and storage media (DNA/RNA Shield, Zymo Research, Irvine, CA, USA, Cat. R1100-250). Saliva samples were used for microRNA analyses. The study participants were requested not to eat, drink, or use any chewing gum an hour before the sampling to avoid any stress factors influencing the production of unstimulated saliva. Saliva sampling was performed prior to the first radiotherapy fraction and immediately after the last fraction.

Isolation was carried out in a class II laminar-flow cabinet to avoid environmental contamination. Total RNA was extracted with the MagMax mirVana total RNA isolation kit (Thermo Fisher Scientific, Waltham, MA, USA) from saliva samples according to the manufacturer’s instructions. We used nuclease-free water (Thermo Fisher Scientific, Waltham, MA, USA) as a negative isolation control (NIC) along with the samples.

RNA quantity was determined using the Qubit™ miRNA Assay Kit (Thermo Fisher Scientific, Waltham, MA, USA) with fluorometric measurement on the Qubit™ 4 Fluorometer. RNA integrity number (RIN) was evaluated via automated electrophoresis using the Agilent 4200 TapeStation System with RNA ScreenTape and RNA ScreenTape Buffer (Agilent Technologies, Santa Clara, CA, USA). All samples had an RIN value above 6. RNA purity was measured using the NanoDrop™ 2000 Spectrophotometer (Thermo Fisher Scientific, Waltham, MA, USA). After quality and quantity measurements, RNA samples were stored at −80 °C until further use.

### 4.3. Library Preparation and Sequencing

Small RNA libraries were prepared using the NEBNext^®^ Small RNA Library Prep Set for Illumina^®^ (New England Biolabs Inc., Ipswich, MA, USA), according to the manufacturer’s instructions. Briefly, 500 ng of total RNA in a 6 μL volume was applied for library preparation. The following steps—including 3′ and 5′ SR adaptor ligation, reverse transcription, primer hybridization, and PCR amplification—were assessed following the kit’s instructions. After PCR amplification, the QIAQuick PCR Purification Kit (Qiagen, Hilden, Germany) in combination with MagSI-NGSPREP Plus beads (Magtivio B.V., Nuth, The Netherlands) were used for cDNA constructs purification according to the manufacturers’ instructions. E-Gel^®^ EX 2% Agarose gels were used with the E-Gel™ Power Snap Electrophoresis Device (Thermo Fisher Scientific, Waltham, MA, USA) for size selection of the amplified cDNA. Finally, 5 μL of a pooled 4 nM DNA library was used for sequencing on the Illumina MiSeq platform. The purified cDNA libraries were measured with Agilent 4200 TapeStation System using D1000 ScreenTape (Agilent Technologies, Santa Clara, CA, USA) and D1000 Sample Buffer (Agilent Technologies, Santa Clara, CA, USA). Finally, a pooled 4 nM RNA library was used for sequencing. The pooled library as well as a standard 1% PhiX Control Library used as an internal control (Illumina, San Diego, CA, USA) were denatured with 0.2 N NaOH. Sequencing was carried out on the Illumina NextSeq 550 Sequencing System (Illumina) with read lengths of 75 base pairs and 3.5 million single-end reads per sample, on average.

### 4.4. Bioinformatic Analyses

Initially, sequencing adapters were trimmed from the demultiplexed library using Cutadapt software (version 4.0), and AGATCGGAAGAGCACACGTCTGAACTCCAGTCAC query sequences were filtered. The FastQC program (Babraham Institute, Cambridge, UK) was used for quality control of the read data. Additional trimming was assessed with the Trimmomatic program [60].

The resulting clean reads were further processed using the StrandNGS software (version 4.1). The alignment was mapped to the human genome, version Hg38. Trimmed mean of M values (TMM) normalization was applied and the calculated read per million (RPM) considering normalization factors was derived with the TMM method (RPM = read counts per miRNA/total read count × 1,000,000/TMM normalizing factor). miRNAs below 10 RPM were excluded from further analyses.

The receiver operating characteristic (ROC) curve for miRNAs validated successfully by RT-qPCR was generated using relative gene expression values and using easyROC (version 1.3.1). The diagnostic potential of the preselected miRNA biomarkers was assessed by plotting true positive rates (sensitivity, y-axis) against false positive rates (1–specificity, x-axis). The area under the ROC curve (AUC) was calculated and applied as an accurate index to evaluate the diagnostic performance of the selected miRNAs. The cutoff expression values were defined to estimate the number of correctly classified samples.

### 4.5. Validation of miRNA-Seq Data by RT-qPCR

Total RNA (2 ng) was used for miRNA-specific reverse transcription using a TaqMan™ Advanced miRNA cDNA Synthesis Kit (Thermo Fisher Scientific). Quantitative real-time PCR with 7 TaqMan™ Gene Expression Assays (Thermo Fisher Scientific) was performed to detect miRNA expression profiles in 3 independent technical replicates, including negative controls (no template from RNA isolation and reverse transcription), using a LightCycler^®^ 480 Real-Time PCR System (Roche Diagnostics, Risch-Rotkreuz, Switzerland). All patient samples were analyzed with quantitative real-time PCR. The expression level of hsa-let-7i-5p was the most stable among all the samples, therefore it was used for normalization. Mean Ct values of technical replicates were used for calculations. As an internal control, has-miR-let7-5p was used. The transcript levels were normalized to the housekeeping gene (hsa-miR-let-7-5p) using the ΔCt method. Subsequently, the data were further normalized to the expression levels observed in the control group (healthy volunteers) following the ΔΔCT method.

### 4.6. Interaction Network Determination from KEGG and miRNA Database

miRabel together with KEGG pathway databases were used to determine the gene targets of significantly altered miRNAs. After gene prediction, we used the Kyoto encyclopedia of genes and genomes (KEGG) database to perform pathway analyses. We evaluated those biological pathways that are relevant in CR pathogenesis (Kanehisa, M. & Goto, S. KEGG: Kyoto encyclopedia of genes and genomes) [61].

### 4.7. Statistical Analysis

Statistical analyses were performed using a GraphPad Prism statistical software package (GraphPad Software, La Jolla, CA, USA, version 8.0). Statistical comparisons between the initial phase and the finishing phase were performed with the Mann–Whitney test. We calculated the statistical differences between healthy volunteers and adenocarcinoma cancer patients using the Mann–Whitney test. Significance was accepted at *p*-value < 0.05. Data are presented as mean ± standard deviation (SD).

## Figures and Tables

**Figure 1 ijms-26-10506-f001:**
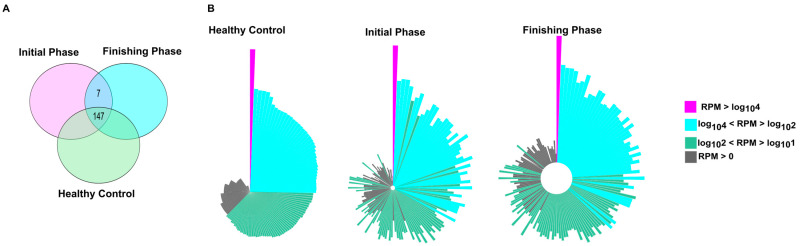
Transcriptomic profiling revealed common and distinct miRNAs among the study participants. (**A**) Venn diagram illustrates the number of miRNAs that were shared and unique between different study groups. (**B**) The normalized distribution patterns of 157 miRNAs in healthy and patient groups, either in the initial or finishing phase, are visualized as circo plots. On the Circo plots, pink, blue, green, and gray correspond to miRNAs with high (RPM > log10^4^), medium (log10^4^ < RPM > log10^2^), low (log10^2^ < RPM > log10^1^), and RPM > 0 read per million values, respectively. In every case, the order of the miRNAs (the representative bars) was the same. Bar lengths refer to the log10 RPM sequence number.

**Figure 2 ijms-26-10506-f002:**
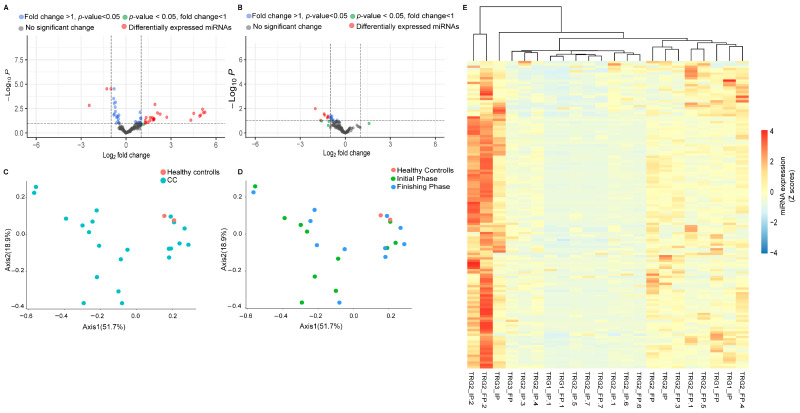
Altered miRNAs expression patterns between the study cohorts. A volcano plot was generated to identify the miRNAs showing fold differences between the rectal cancer patients and healthy volunteers (**A**) as well as between the initial and the finishing phases (**B**) with statistical significance (*p* values ≤ 0.05) and expressing log2-fold changes greater than 1 and lower than −1 using the LIMMA statistical model. Each dot represents a single miRNA. The dashed line on the y-axis indicates the *p*-value = 0.05 threshold with statistically significant (*p* < 0.05) gene expression (up- and downregulation, respectively). Red circles indicate differentially expressed miRNas with statistically significant overexpression and underexpression. We calculated the statistical differences between healthy volunteers and adenocarcinoma cancer patients using the Mann–Whitney test. Ordination pattern analyses representing the differences in distribution patterns between colon cancer (CC) (blue) and healthy volunteers (red) (**C**). The differences between healthy control (red), initial phase (green), and finishing phase (blue) (**D**) were also assessed. Cluster heatmap of 154 miRNAs with RPM ≥ 10 (**E**).

**Figure 3 ijms-26-10506-f003:**
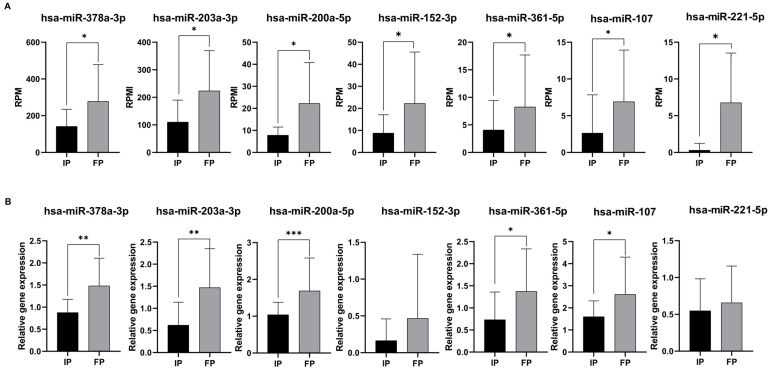
Validation of differentially expressed miRNAs by RT-qPCR. Data shows small RNA sequencing results (**A**). Data shows RPM (read per million vales). Normalized relative gene expressions (normalized to hsa-let-7i-5p) of miRNAs were calculated with the Livac formula (**B**). As an internal control, has-miR-let7-5p was used. The amount of the transcripts was normalized to those of the housekeeping gene (has-miR-let7-5p) using the ΔCT method. Finally, the results were further normalized to the expression of the control (healthy volunteers) (ΔΔCT method). Healthy volunteers’ (control) relative gene expressions were 1 in every case and are not shown in the figure. Statistical comparison between the initial phase and finishing phase was performed with the Mann–Whitney test. Data is shown as mean ± SD. Asterisks report statistical significance * *p* < 0.05; ** *p* < 0.005; and *** *p* < 0.001. IP: initial phase, FP: finishing phase, and RPM: read per million.

**Figure 4 ijms-26-10506-f004:**
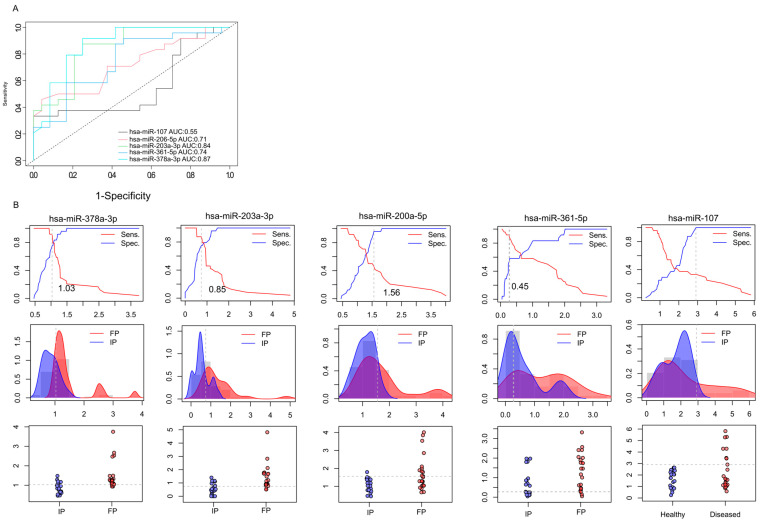
Prognostic performance of miRNAs. ROC (receiver operating characteristics) analyses illustrate the diagnostic performance of the 5 selected microRNA at different discriminatory thresholds (**A**). Data shows the differences between the initial phase (IP) and the finishing phase (FP). Line graphs were used to calculate the optimal cutoff points (**B**). Discriminatory power graphs represent the distribution of the relative gene expression values in IP and FP.

**Figure 5 ijms-26-10506-f005:**
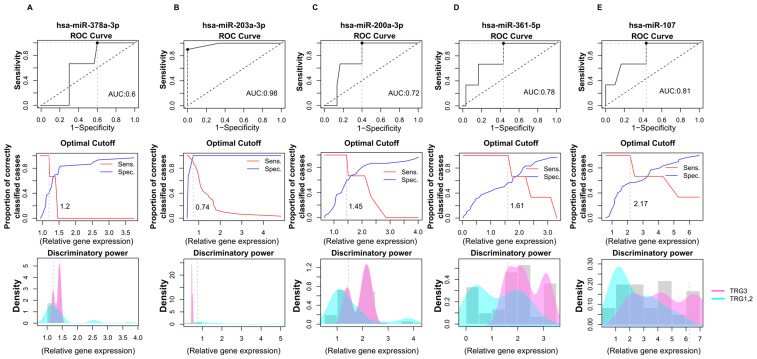
Accuracy for predicting tumor regression grade based on the relative gene expression of significantly altered miRNAs. ROC (receiver operating characteristics) analyses were assessed based on miRNAs expressions determined by RT-qPCR. The following miRNAs’ discriminatory powers were determined for this analysis: hsa-miR-378a-3p (**A**), hsa-miR-203a-3p (**B**), hsa-miR-200a-5p (**C**), hsa-miR-361-5p (**D**), and hsa-miR-107p (**E**). Data shows the differences in miRNAs expressions between the TRG1 and TRG2 groups compared to the TRG3 tumor regression patients’ group. Line graphs were used to calculate the optimal cutoff points. Discriminatory power graphs represent the distribution of the relative gene expression values in TRG1 with TRG2 and TRG3. TRG: tumor regression grade.

**Figure 6 ijms-26-10506-f006:**
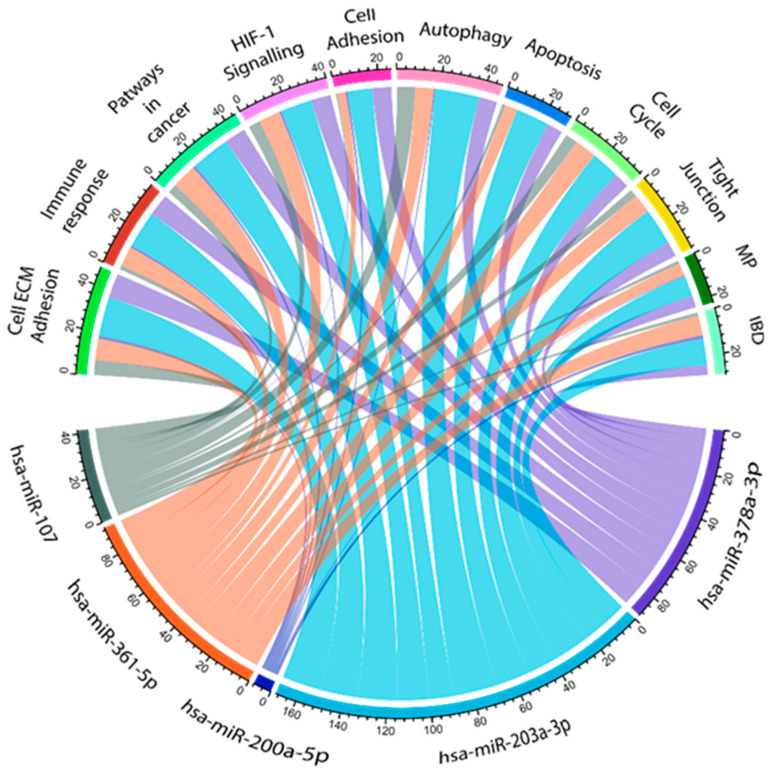
Interaction network of CR-related significantly dysregulated miRNAs. Information assessed from the KEGG and miRabel databases revealed 11 possible biological pathways that can be related to the investigated miRNAs. ECM: extracellular matrix, MP: metabolic pathways, and IBD: inflammatory bowel disease. Cord plot was generated in R studio (ver 401) with circlize pipeline.

**Table 1 ijms-26-10506-t001:** Study cohort; patients’ demographics. The table represents the individual clinical characteristics of the research participants. (T: Tumor size, N: lymph node involvement, y: after neoadjuvant therapy, p: pathological, G: grade, and TRG: tumor regression grade, T3b: tumor extends 1–5 mm beyond muscularis propria (mild invasion), T3c: tumor extends 5–15 mm beyond muscularis propria (moderate invasion), N1a: metastasis in 1 regional lymph node).

Patient ID	Gender	Age	Initial Clinical Stage	After Neoadjuvant Therapy Clinical Stage	After Surgery Pathological Stage	Grade	Modified Ryan Scheme TRG	Localization
58511	Female	66	T3bN2	T2N1	ypT3ypN1a	G1	TRG 1	upper
66921	Male	68	T3cN2	T3bN0	ypT3ypN0	G2	TRG 1	middle
66935	Female	62	T3cN1	T3bN0	ypT3ypN1a	G2	TRG 3	middle
67028	Male	62	T1N1	T1N1	ypT1ypN0	G2	TRG 2	lower
67035	Male	50	T2N1	T1N0	ypT2ypN0	G2	TRG 2	lower
67558	Male	70	T4N2	T2N0	ypT2ypN0	G2	TRG 2	lower
67594	Female	58	T3bN1	T3bN0	ypT3ypN0	G2	TRG 2	middle
67673	Female	70	T3bN2	T3bN1	ypT3ypN1a	G2	TRG 2	upper
67858	Male	70	T3cN2	T2N0	ypT2ypN0	G2	TRG 2	lower
67892	Male	71	T3cN2	T3bN0	ypT3ypN0	G2	TRG 2	lower
67898	Female	72	T3bN2	T3bN0	yPT3ypN0	G2	TRG 2	middle

**Table 2 ijms-26-10506-t002:** Demographic and clinical characteristics of the patients (G: grade, T: tumor size, N: node involvement, c: clinical, and TRG: tumor regression grade).

Clinical Parameters (Patients *n* = 11)		
Gender, *n* (%)		
	Male	6 (54.54%)
	Female	5 (45.46%)
Age, median		68
G1, *n* (%)		1 (9.09%)
G2, *n* (%)		10 (90.91%)
Location, *n* (%)		
	upper	2 (18.18%)
	middle	4 (36.36%)
	lower	5 (45.46%)
cT, *n* (%)		
	1	1 (9.09%)
	2	1 (9.09%)
	3	8 (72.73%)
	4	1 (9.09%)
cN, *n* (%)		
	1	4 (36.36%)
	2	7 (63.64%)
TRG (modified Ryan scheme), *n* (%)		
	1	2 (18.18%)
	2	8 (72.73%)
	3	1 (9.09%)

## Data Availability

All sequence data used in the analyses were deposited in the Sequence Read Archive (SRA) (http://www.ncbi.nlm.nih.gov/sra) under PRJNA1321322 (8 September 2025). Until publication, please use the reviewers’ link: https://dataview.ncbi.nlm.nih.gov/object/PRJNA1321322?reviewer=lkepua0rvaqu5m3d29og63ualg (accessed on 8 September 2025).

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
