# Peer review of "Analysis of the Circulating miRNome Expression Profile in Saliva Samples After Neoadjuvant Chemoradiotherapy in a Rectal Cancer Study Population Using Next-Generation Sequencing"

_ijms, 2025, doi:10.3390/ijms262110506_

Round 1

Reviewer 1 Report

Comments and Suggestions for Authors

The article makes good mention of the miRs that could be modified in colorectal cancer patients, as well as the differences between those who respond to treatment and those who do not. It is an interesting study with several notable findings. However, I have some concerns that could be incorporated into the manuscript to improve it.
Introduction:
-I believe it is important to mention the existing evidence regarding differences in miR expression in people who respond to treatments for other cancers or other pathologies. This is to further support the objective of the study.
-All abbreviations that appear should be defined in the order in which they appear.
Results:
-The description of Table 1 repeats the word "table 1."
-In the Venn diagram, it would be important to mention how many miRs were differentially expressed in the initial phase, how many in the final phase, and how many in the control group, as mentioned in the text.
-In the description of the results, it would be easier to follow the text if you clearly indicated whether they were over- or underexpressed.
-Figure 3 needs further description, including the meaning of the abbreviations and titles appropriate to sections a and b.
-Figure 6 could be clearer about the program used to create the figure.
Discussion
-There are extra spaces that should be removed, and there are also references that should be "joined," for example, line 424.
-There are spelling errors, such as "has" instead of "hsa."
-Again, I emphasize being more specific about overexpression to understand it, as well as the genes that these miRs specifically regulate and how this may be affecting disease progression.
-In one section, you refer to two miRs that increased in saliva, but you don't mention which group or context.
Methodology
-You should mention how you performed the second delta, since the results in Figure 3 show what appears to be expression at a single delta. You should be more clear.
-Finally, they left the "funding" and "acknowledgments" fields blank; if there aren't any, they should indicate that they don't exist.

Author Response

Review 1

Thank you for your thorough comments and suggestions regarding our manuscript. We have responded to the points raised in detail below and made the appropriate correction/rewording in the manuscript, which are highlighted in yellow in the returned document.

Introduction:
1) I believe it is important to mention the existing evidence regarding differences in miR expression in people who respond to treatments for other cancers or other pathologies. This is to further support the objective of the study.

A: We added the following extra part about the existing evidence regarding differences in miR expression in people who respond to treatments for other cancers or other pathologies to the introduction line 75-97

In recent years, considerable attention has been directed toward alterations in microRNAs in response to cancer therapy. In breast cancer, Lindholm et al. identi-fied miR-4465 as a potential biomarker during neoadjuvant bevacizumab and chemotherapy, showing that its downregulation effectively predicted reduced pro-liferation (17). Based on the in vitro esophageal cancer study by Hummel et al., dysregulation of miR-27b-3p, miR-193b-3p, miR-192-5p, miR-378a-3p, miR-125a-5p, and miR-18a-3p was observed, consistent with their role in the negative post-transcriptional regulation of KRAS, TYMS, ABCC3, CBL-B, and ERBB2 expres-sion (18). Yu et al. demonstrated that treatment of colorectal cancer cells with 5-FU led to the upregulation of 17 miRNAs (miR‐19a, miR‐20, miR‐21, miR‐23a, miR‐25, miR‐27a/b, miR‐29a, miR‐30e, miR‐124b, miR‐132, miR‐133a, miR‐141, miR‐147, miR‐151, miR‐182, miR‐185) and the downregulation of three others (miR‐200b, miR‐210, miR‐224) (19). Moreover, mir-21 was able to predict incomplete response in rectal adenocarcinoma after chemoradiotherapy (20). According to Valenzuela et al., the miR-200 family is capable of reversing EMT phenotypes, thereby restoring chemosensitivity. Furthermore, miRNAs such as miR-19a and miR-625-3p have shown predictive value regarding chemotherapy response. Supporting this, Badr et al. reported that alterations in miR-19a-3p provide important insights into the long-term effects of 5-FU in colorectal cancer, particularly in relation to recurrence and disease progression (21). Conversely, Chun-Ming Huan et al. found that miR-148a promotes apoptosis under chemoradiation while simultaneously sup-pressing proliferation in colorectal cancer cells. According to previous results, mir-345 was associated with unfavorable chemoradiotherapy response and poor lo-coregional control in rectal adenocarcinoma (22).

2) All abbreviations that appear should be defined in the order in which they appear.

A: All abbreviations are defined at their first occurrence in the text as requested.

Results:
3) The description of Table 1 repeats the word "table 1."

A: We have corrected the repetition in the description of Table 1.

4) In the Venn diagram, it would be important to mention how many miRs were differentially expressed in the initial phase, how many in the final phase, and how many in the control group, as mentioned in the text.

A: We added the following sentences in the text line 163-1164 as requested

  • Interestingly, from these 7 miRNAs only the hsa-miR-221-5 was significantly upregulated in the finishing phase compared to the initial phase.

5) In the description of the results, it would be easier to follow the text if you clearly indicated whether they were over- or underexpressed.

A: We have revised the Results section to clearly indicate whether each gene or protein was overexpressed or underexpressed, thereby improving the readability and clarity of the text.

6) Figure 3 needs further description, including the meaning of the abbreviations and titles appropriate to sections a and b.

A: We corrected Figure 3 description as requested

7) Figure 6 could be clearer about the program used to create the figure.

A: We have clarified in the figure legend 6 with the following sentence “Cord plot was generated in R studio (ver 401) with circlize pipeline.”

Discussion

8) There are extra spaces that should be removed, and there are also references that should be "joined," for example, line 424.

A: We have carefully reviewed the manuscript and removed extra spaces. References that should be joined (e.g., line 424) have also been corrected

9) There are spelling errors, such as "has" instead of "hsa."

A: We have corrected the typographical errors throughout the manuscript, including replacing “has” with “hsa” where appropriate.

10) Again, I emphasize being more specific about overexpression to understand it, as well as the genes that these miRs specifically regulate and how this may be affecting disease progression.

A: We have revised the manuscript to provide more specific details regarding the overexpression of the identified miRNAs. We have supplemented the relevant parts of the discussion with the target genes of the studied miRNAs and how they may influence the prognosis of the disease.

We added the following sentences extra sentences line (326-33) of results part

  • Based on miRabel together with KEGG pathway, we found that hsa-miR-378-3p, hsa-miR-203a-3p and has-miR-361-5p regulate genes that are highly involved in cell ECM adhesion (fibronectin, MMPs, TIMP), Immune response (Caspase-9, MALAT1 and NEAT1) and apoptosis (KLK4, FOXD1-MYC-ODC1, STAT3). hsa-miR-200a-5p mostly influenced those genes that are involved in autophagy (PDK1, Akt/mTOR, ULK1) and cell adhesion (MMP3, MINK1, LAMA4) pathways. Furthermore, hsa-miR-107 (ATG12, LC3-II, HMGB1) mainly exacerbated autophagy mechanism, apoptosis (BCL2, PTEN/AKT) and immune response (SOCS3 / STAT3) pathways.

11) In one section, you refer to two miRs that increased in saliva, but you don't mention which group or context.

A: We have revised the text to specify the group and experimental context in which the two miRNAs were increased in saliva, ensuring the description is clear and unambiguous.

  • During our studies, the expression of two miRNAs was increased in saliva of CRC patients (hsa-miR-152-3p and hsa-miR-221-5p), which we could not validate by PCR analysis.

Methodology
12) You should mention how you performed the second delta, since the results in Figure 3 show what appears to be expression at a single delta. You should be more clear.

A: To improve clarity, we have revised the Materials and Methods section to specify how the second Δ (delta) was calculated in the ΔΔCt method. We have also clarified in the Figure 3 legend that the data represent relative expression levels calculated using this approach.

We have added the following sentences to the Figure 3 legend (lines 246–249).

  • As an internal control glyceraldehyde-3-phosphate dehydrogenase (GAPDH) was used. The amount of the transcripts was normalized to those of the housekeeping gene (GAPDH) using the ΔCT method. Finally, the results were further normalized to the expression of the control (healthy volunteers) (ΔΔCT method).

We have added the following sentences to the Materials and methods (lines 672–675).

  • The transcript levels were normalized to the housekeeping gene (hsa-miR-let-7-5p) using the ΔCt method. Subsequently, the data were further normalized to the expression levels observed in the control group (healthy volunteers) following the ΔΔCt method.

13) Finally, they left the "funding" and "acknowledgments" fields blank; if there aren't any, they should indicate that they don't exist.

A: This research did not receive any specific grant from funding agencies in the public, commercial, or not-for-profit sectors

Reviewer 2 Report

Comments and Suggestions for Authors

Your study explores an innovative and clinically relevant topic: the potential role of salivary microRNAs as non-invasive biomarkers for monitoring treatment response in rectal cancer. The methodology is rigorous, with a strong design that combines next-generation sequencing, RT-qPCR validation, and bioinformatic analyses. The manuscript is well-structured and clearly written, and the methodological approach is thoughtful. However, several important issues need to be addressed to strengthen the work:

  1. Please clarify patient inclusion and exclusion criteria, including comorbidities that may have influenced salivary miRNA expression.
  2. Specify the timing of saliva sample collection in relation to neoadjuvant treatment: Was it obtained immediately after the completion of radiochemotherapy, at a later time point, or at the time of imaging reevaluation?
  3. Please indicate when the imaging reevaluation was performed to assess therapeutic response and after how many weeks of treatment.
  4. The sample size is very small (n = 11), with only one non-responder. This greatly limits statistical power, particularly for ROC analyses, which may appear overstated under these conditions. Presenting the study as exploratory and hypothesis-generating would be more appropriate.
  5. Only 5 of the 7 candidate miRNAs were validated by RT-qPCR. The lack of validation for hsa-miR-152-3p and hsa-miR-221-5p should be critically discussed, including possible technical or biological explanations.
  6. Please clarify whether corrections for multiple testing were applied.
  7. ROC analyses based on such small and unbalanced groups should be interpreted very cautiously. Consider reframing these findings as preliminary observations rather than definitive results.
  8. When discussing biomarker potential, emphasize the necessity for external validation in larger and independent cohorts.
  9. The conclusion should be reframed to emphasize that the findings are preliminary, requiring validation in larger, multi-center cohorts. 

Author Response

Review 2

Thank you for your thorough comments and suggestions regarding our manuscript. We have responded to the points raised in detail below and made the appropriate correction/rewording in the manuscript, which are highlighted in green in the returned document.

  1. Please clarify patient inclusion and exclusion criteria, including comorbidities that may have influenced salivary miRNA expression.

A: We added the following extra sentences line 535-552 of the materials and methods     part.

- In addition to a diagnosis of locally advanced rectal adenocarcinoma, adequate patient compliance and provision of voluntary, informed consent were mandatory inclusion criteria. Age and sex were not considered determining factors for patient inclusion.

Exclusion criteria encompassed all physiological conditions that contraindicated either chemotherapy or radiotherapy, as well as comorbidities with the potential to signifi-cantly influence miRNA expression. Consequently, only patients with an overall condi-tion suitable for neoadjuvant chemoradiotherapy (ECOG performance status 0–2) were considered eligible. Further exclusion criteria included the presence of any additional malignant disease, a history of radiotherapy to any anatomical region, or prior chemo-therapy or the presence of distant metastases. None of the patients presented with ac-tive oral or systemic infections, immunological disorders, or contraindications such as bone marrow failure, anemia, leukopenia, thrombocytopenia, untreated hepatic or renal impairment, or cardiac insufficiency. Patients with regular alcohol or drug consumption or active smoking habits were also excluded.

With respect to comorbidities, essential hypertension was the most common, observed in 8 patients (72.7%). In addition, one patient had non–insulin-dependent diabetes mellitus, one suffered from tachycardia, and another had a history of nephrolithiasis.

  1. Specify the timing of saliva sample collection in relation to neoadjuvant treatment: Was it obtained immediately after the completion of radiochemotherapy, at a later time point, or at the time of imaging reevaluation?

A: We added the following extra sentences line 580-582 of the materials and methods part.

  • Imaging reevaluation was performed within one week after the last cycle of consolidation chemotherapy.

  1. Please indicate when the imaging reevaluation was performed to assess therapeutic response and after how many weeks of treatment.

A: We added the following extra sentences line 610-611 of the materials and methods part.

  • Saliva sampling was performed prior to the first radiotherapy fraction and immediately after the last fraction.

  1. The sample size is very small (n = 11), with only one non-responder. This greatly limits statistical power, particularly for ROC analyses, which may appear overstated under these conditions. Presenting the study as exploratory and hypothesis-generating would be more appropriate.

A: Accordingly, we have revised the manuscript to present the study as exploratory and hypothesis-generating, emphasizing the preliminary nature of the findings.

We added the following sentences in result figure 4-part line 266-268.

  • Based on our preliminary results, hsa-miR-378a-3p, hsa-miR-203a-3p hsa-miR-200a-5p and hsa-miR-361 have a promising potential to discriminate the initial phase from the finishing phase but further validations are needed.

We have added the following sentences to the Figure 5 legend (lines 297-302).

  • This result suggests that hsa-miR-203a-3p in combination with hsa-miR-200a-3p and hsa-miR-361 could be a putative predictive marker for tumor regression but further validations are needed.
  1. Only 5 of the 7 candidate miRNAs were validated by RT-qPCR. The lack of validation for hsa-miR-152-3p and hsa-miR-221-5p should be critically discussed, including possible technical or biological explanations.

A: We validated all of the 7 candidate miRNS, but only in the case of 5 miRNAs were found significant differences caused by radiotherapy.

Please see the following lines 232-237:

Among these five miRNAs all of them were up-regulated compared to the initial phase. As a result, 5 significantly altered miRNAs (hsa-miR-378a-3p, hsa-miR-203a-3p, hsa-miR-200a-5p, hsa-miR-361-5p and hsa-miR-107p) were identified as a candidate for further testing. In the case of hsa-miR-152-3p and hsa-miR-221-5p we observed similar expression patterns with the RNA sequencing results, nevertheless the expression differences were not significant between the initial and the finishing phase.

  1. Please clarify whether corrections for multiple testing were applied.

A: We appreciate the reviewer’s observation. All statistical analyses involved comparisons between two groups using the Mann–Whitney test. Because each comparison was performed individually, corrections for multiple testing were not applied. This has been clarified in the revised manuscript line 685-689. We have also supplemented the figure legends with the relevant statistical information

  • Statistical comparison between the initial phase and finishing phase were performed with Mann Whitney test. We calculated the statistical differences between healthy volunteers and adenocarcinoma cancer patients using Mann Whitney test. Significance was accepted at p-value < 0.05. Data are presented as mean±standard deviation (SD).
  1. ROC analyses based on such small and unbalanced groups should be interpreted very cautiously. Consider reframing these findings as preliminary observations rather than definitive results.

A: Given the small sample size and the limited number of non-responders, we recognize that these results have limited statistical power. The manuscript has been revised to clearly frame the ROC findings as preliminary and exploratory, emphasizing the need for validation in larger cohorts.

We added the following sentences in result figure 4-part line 265-267.

  • Based on our preliminary results, hsa-miR-378a-3p, hsa-miR-203a-3p hsa-miR-200a-5p and hsa-miR-361 have a promising potential to discriminate the initial phase from the finishing phase but further validations are needed.

We have added the following sentences to the Figure 5 (lines 297-302).

  • This result suggests that hsa-miR-203a-3p in combination with hsa-miR-200a-3p and hsa-miR-361 could be a putative predictive marker for tumor regression but further validations are needed. It is important to note that our findings are preliminary observations and because of the small sample and unbalanced sample size external validations in larger, independent cohorts to ensure reproducibility and generalizability of the findings are needed.
  1. When discussing biomarker potential, emphasize the necessity for external validation in larger and independent cohorts.

A: In accordance with the suggestion, we have revised the Discussion to emphasize the importance of external validation in larger and independent cohorts to confirm the robustness and generalizability of the proposed biomarkers line 205-508.

  • Based on our findings, we suggest that hsa-miR-203a-3p in combination with hsa-miR-200a-3p and hsa-miR-361 could be a putative predictive marker for tumor regression but further validations in a larger and independent cohort to confirm the robustness and generalizability of the proposed biomarkers are needed.
  1. The conclusion should be reframed to emphasize that the findings are preliminary, requiring validation in larger, multi-center cohorts. 

We have added the following sentences to discussion lines 514-524.

  • Our study has several limitations. First the aforementioned low and unbalanced sample size might have resulted, that we got excellent diagnostic potential in the case of hsa-miR-203a-3p, hsa-miR-200a-3p and hsa-miR-361. We acknowledge that the relatively small sample size may have influenced the ROC values; therefore, with a larger cohort, these values might be lower than those observed in the present study. Second, miRNAs have a pleiotropic effect, they can impact their gen regulatory impacts on several biological pathways. Therefore, it is difficult to determine which biological pathways were influenced by the altered expression of the proposed miRNAs and how these changes relate to treatment response following neoadjuvant chemoradiotherapy. Continued research and validation are essential draw far-reaching conclusion whether and how miRNAs contribute to CR pathology and to support their differential diagnostic potential.

Round 2

Reviewer 1 Report

Comments and Suggestions for Authors

I recommend removing the "bold" from the regular text, as well as italicizing the gene names.
Otherwise, you have significantly improved the manuscript. Congratulations.

Author Response

Thank you very much for your feedback and kind words. we romoved the bold formatting from the regular text and italicized the gene names as suggested. I really appreciate your guidance and am glad to hear that the manuscript has improved.

Reviewer 2 Report

Comments and Suggestions for Authors

Thank you to the authors for their clear and precise point-by-point responses to my comments.
However, it appears that the answers to the following two questions may have been inadvertently switched.

Specify the timing of saliva sample collection in relation to neoadjuvant treatment: Was it obtained immediately after the completion of radiochemotherapy, at a later time point, or at the time of imaging reevaluation?
A: We added the following extra sentences line 580-582 of the materials and methods part.
•    Imaging reevaluation was performed within one week after the last cycle of consolidation chemotherapy.

1.    Please indicate when the imaging reevaluation was performed to assess therapeutic response and after how many weeks of treatment.
A: We added the following extra sentences line 610-611 of the materials and methods part.
•    Saliva sampling was performed prior to the first radiotherapy fraction and immediately after the last fraction.

Author Response

Thank you for pointing this out. We apologize for the confusion. We uploaded the corrected correspondence between the questions and their respective answers. We would like to thank you for their valuable comments and constructive feedback, which have helped us improve the quality of our manuscript.

Review 2

Thank you for your thorough comments and suggestions regarding our manuscript. We have responded to the points raised in detail below and made the appropriate correction/rewording in the manuscript, which are highlighted in green in the returned document.

  1. Please clarify patient inclusion and exclusion criteria, including comorbidities that may have influenced salivary miRNA expression.

A: We added the following extra sentences line 535-552 of the materials and methods     part.

- In addition to a diagnosis of locally advanced rectal adenocarcinoma, adequate patient compliance and provision of voluntary, informed consent were mandatory inclusion criteria. Age and sex were not considered determining factors for patient inclusion.

Exclusion criteria encompassed all physiological conditions that contraindicated either chemotherapy or radiotherapy, as well as comorbidities with the potential to signifi-cantly influence miRNA expression. Consequently, only patients with an overall condi-tion suitable for neoadjuvant chemoradiotherapy (ECOG performance status 0–2) were considered eligible. Further exclusion criteria included the presence of any additional malignant disease, a history of radiotherapy to any anatomical region, or prior chemo-therapy or the presence of distant metastases. None of the patients presented with ac-tive oral or systemic infections, immunological disorders, or contraindications such as bone marrow failure, anemia, leukopenia, thrombocytopenia, untreated hepatic or renal impairment, or cardiac insufficiency. Patients with regular alcohol or drug consumption or active smoking habits were also excluded.

With respect to comorbidities, essential hypertension was the most common, observed in 8 patients (72.7%). In addition, one patient had non–insulin-dependent diabetes mellitus, one suffered from tachycardia, and another had a history of nephrolithiasis.

  1. Specify the timing of saliva sample collection in relation to neoadjuvant treatment: Was it obtained immediately after the completion of radiochemotherapy, at a later time point, or at the time of imaging reevaluation?

A: We added the following extra sentences of the materials and methods part.

  • Saliva sampling was performed prior to the first radiotherapy fraction and immediately after the last fraction.

  1. Please indicate when the imaging reevaluation was performed to assess therapeutic response and after how many weeks of treatment.

A: We added the following extra sentences of the materials and methods part.

  • Imaging reevaluation was performed within one week after the last cycle of consolidation chemotherapy.

  1. The sample size is very small (n = 11), with only one non-responder. This greatly limits statistical power, particularly for ROC analyses, which may appear overstated under these conditions. Presenting the study as exploratory and hypothesis-generating would be more appropriate.

A: Accordingly, we have revised the manuscript to present the study as exploratory and hypothesis-generating, emphasizing the preliminary nature of the findings.

We added the following sentences in result figure 4-part line 266-268.

  • Based on our preliminary results, hsa-miR-378a-3p, hsa-miR-203a-3p hsa-miR-200a-5p and hsa-miR-361 have a promising potential to discriminate the initial phase from the finishing phase but further validations are needed.

We have added the following sentences to the Figure 5 legend (lines 297-302).

  • This result suggests that hsa-miR-203a-3p in combination with hsa-miR-200a-3p and hsa-miR-361 could be a putative predictive marker for tumor regression but further validations are needed.
  1. Only 5 of the 7 candidate miRNAs were validated by RT-qPCR. The lack of validation for hsa-miR-152-3p and hsa-miR-221-5p should be critically discussed, including possible technical or biological explanations.

A: We validated all of the 7 candidate miRNS, but only in the case of 5 miRNAs were found significant differences caused by radiotherapy.

Please see the following lines 232-237:

Among these five miRNAs all of them were up-regulated compared to the initial phase. As a result, 5 significantly altered miRNAs (hsa-miR-378a-3p, hsa-miR-203a-3p, hsa-miR-200a-5p, hsa-miR-361-5p and hsa-miR-107p) were identified as a candidate for further testing. In the case of hsa-miR-152-3p and hsa-miR-221-5p we observed similar expression patterns with the RNA sequencing results, nevertheless the expression differences were not significant between the initial and the finishing phase.

  1. Please clarify whether corrections for multiple testing were applied.

A: We appreciate the reviewer’s observation. All statistical analyses involved comparisons between two groups using the Mann–Whitney test. Because each comparison was performed individually, corrections for multiple testing were not applied. This has been clarified in the revised manuscript line 685-689. We have also supplemented the figure legends with the relevant statistical information

  • Statistical comparison between the initial phase and finishing phase were performed with Mann Whitney test. We calculated the statistical differences between healthy volunteers and adenocarcinoma cancer patients using Mann Whitney test. Significance was accepted at p-value < 0.05. Data are presented as mean±standard deviation (SD).
  1. ROC analyses based on such small and unbalanced groups should be interpreted very cautiously. Consider reframing these findings as preliminary observations rather than definitive results.

A: Given the small sample size and the limited number of non-responders, we recognize that these results have limited statistical power. The manuscript has been revised to clearly frame the ROC findings as preliminary and exploratory, emphasizing the need for validation in larger cohorts.

We added the following sentences in result figure 4-part line 265-267.

  • Based on our preliminary results, hsa-miR-378a-3p, hsa-miR-203a-3p hsa-miR-200a-5p and hsa-miR-361 have a promising potential to discriminate the initial phase from the finishing phase but further validations are needed.

We have added the following sentences to the Figure 5 (lines 297-302).

  • This result suggests that hsa-miR-203a-3p in combination with hsa-miR-200a-3p and hsa-miR-361 could be a putative predictive marker for tumor regression but further validations are needed. It is important to note that our findings are preliminary observations and because of the small sample and unbalanced sample size external validations in larger, independent cohorts to ensure reproducibility and generalizability of the findings are needed.
  1. When discussing biomarker potential, emphasize the necessity for external validation in larger and independent cohorts.

A: In accordance with the suggestion, we have revised the Discussion to emphasize the importance of external validation in larger and independent cohorts to confirm the robustness and generalizability of the proposed biomarkers line 205-508.

  • Based on our findings, we suggest that hsa-miR-203a-3p in combination with hsa-miR-200a-3p and hsa-miR-361 could be a putative predictive marker for tumor regression but further validations in a larger and independent cohort to confirm the robustness and generalizability of the proposed biomarkers are needed.
  1. The conclusion should be reframed to emphasize that the findings are preliminary, requiring validation in larger, multi-center cohorts. 

We have added the following sentences to discussion lines 514-524.

  • Our study has several limitations. First the aforementioned low and unbalanced sample size might have resulted, that we got excellent diagnostic potential in the case of hsa-miR-203a-3p, hsa-miR-200a-3p and hsa-miR-361. We acknowledge that the relatively small sample size may have influenced the ROC values; therefore, with a larger cohort, these values might be lower than those observed in the present study. Second, miRNAs have a pleiotropic effect, they can impact their gen regulatory impacts on several biological pathways. Therefore, it is difficult to determine which biological pathways were influenced by the altered expression of the proposed miRNAs and how these changes relate to treatment response following neoadjuvant chemoradiotherapy. Continued research and validation are essential draw far-reaching conclusion whether and how miRNAs contribute to CR pathology and to support their differential diagnostic potential.